# Design of an Ultrasound Sensing System for Estimation of the Porosity of Agricultural Soils

**DOI:** 10.3390/s24072266

**Published:** 2024-04-02

**Authors:** Stuart Bradley, Chandra Ghimire

**Affiliations:** 1Inverse Acoustics Ltd., 73 Daffodil Street, Auckland 0604, New Zealand; 2AgResearch, Lincoln Research Centre, Private Bag 4749, Christchurch 8140, New Zealand; chandra.ghimire@scionresearch.com; 3Scion, 10 Kyle Street, Riccarton, Christchurch 8011, New Zealand

**Keywords:** soil porosity, ultrasound, ultrasonic arrays, reflected ultrasound, specular and diffuse ultrasound reflections

## Abstract

The design of a readily useable technology for routine paddock-scale soil porosity estimation is described. The method is non-contact (proximal) and typically from “on-the-go” sensors mounted on a small farm vehicle around 1 m above the soil surface. This ultrasonic sensing method is unique in providing estimates of porosity by a non-invasive, cost-effective, and relatively simple method. Challenges arise from the need to have a compact low-power rigid structure and to allow for pasture cover and surface roughness. The high-frequency regime for acoustic reflections from a porous material is a function of the porosity *ϕ*, the tortuosity *α*_∞_, and the angle of incidence *θ*. There is no dependence on frequency, so measurements must be conducted at two or more angles of incidence *θ* to obtain two or more equations in the unknown soil properties *ϕ* and *α*_∞_. Sensing and correcting for scattering of ultrasound from a rough soil surface requires measurements at three or more angles of incidence. A system requiring a single transmitter/receiver pair to be moved from one angle to another is not viable for rapid sampling. Therefore, the design includes at least three transmitter/reflector pairs placed at identical distances from the ground so that they would respond identically to power reflected from a perfectly reflecting surface. A single 25 kHz frequency is a compromise which allows for the frequency-dependent signal loss from a natural rough agricultural soil surface. Multiple-transmitter and multiple-microphone arrays are described which give a good signal-to-noise ratio while maintaining a compact system design. The resulting arrays have a diameter of 100 mm. Pulsed ultrasound is used so that the reflected sound can be separated from sound travelling directly through the air horizontally from transmitter to receiver. The average porosity estimated for soil samples in the laboratory and in the field is found to be within around 0.04 of the porosity measured independently. This level of variation is consistent with uncertainties in setting the angle of incidence, although assumptions made in modelling the interaction of ultrasound with the rough surface no doubt also contribute. Although the method is applicable to all soil types, the current design has only been tested on dry, vegetation-free soils for which the sampled area does not contain large animal footprints or rocks.

## 1. Introduction

Soil porosity, or the ratio of the void volume to the total volume of a soil sample, is an important factor in maintaining good soil health [1,2,3]. Measuring soil porosity aids in assessing if current land use and practices are sustaining soil health and limiting impacts on the environment and food production. Traditional methods for measuring soil porosity include measuring the extra mass of water which is required to saturate a soil sample; use of a pycnometer to measure the air volume in the pore space for a soil sample in a gas-tight chamber; and compression of a soil sample to estimate the solid volume of soil [4,5]. These are all direct methods requiring soil samples. Proximal soil sensing refers to field-based techniques that can measure soil properties from 2 m or less above the soil surface [6]. Such sensing technologies have the potential to generate vastly more data at lower cost, providing that the complexity of such technologies does not present a barrier to adoption by land managers such as farmers [7]. Many proximal soil sensing techniques exist, including ion-sensitive field effect transistors to measure soil pH and soil nutrients; visible-near-infrared spectrometers to measure organic carbon content and mineral composition; γ-ray sensors of soil spatial variability and mineral content; X-ray fluorescence and laser-induced breakdown spectroscopy to measure soil elemental analysis; ground-penetrating radar to measure soil water content; and electromagnetic induction to measure soil electrical conductivity [8,9,10,11]. Future extensions of magnetic resonance sounding may allow soil porosity to be estimated [9], and it has been suggested that a combination of ground-penetrating radar and seismic data could allow both soil moisture and soil porosity to be estimated [12], but such methods have yet to be proven. Here, we concentrate on non-invasive (non-contact) proximal sensing to obtain soil porosity; currently there are no methods to do this.

A new method is described for estimating the porosity of natural agricultural soils based on reflections of ultrasound. This technology can potentially be applied at the paddock or farm scale, based on sensors mounted on a small farm vehicle around 1 m above the soil surface. Analysis of several soil samples having different degrees of surface roughness shows that soil porosity can be estimated to within around 0.04.

The technical details of the design are described here. In Methods (Section 2), the key design parameters are examined. These are the geometry of the sensor packages, the operating acoustic frequency, the acoustic pulse design, the transmitter and receiver array diameters, the angles of incidence (determined by the lateral displacement of transmitter and receiver arrays), and the consequential “footprint” on the soil surface. The hardware implementation based on these parameters is covered, including calibration, gains, and signal-to-noise performance. The overall performance of this design is discussed in the Results (Section 3) with reference to measurements on soil samples in the laboratory and field. Section 4 is a discussion of the design outcomes and potential operational use on farms, followed by a short Conclusions section (Section 5).

## 2. Materials and Methods

### 2.1. Transmitter and Receiver Placement

An acoustic transmitter is directed downward at an azimuth angle *θ* towards a point on the soil surface, which is a distance *r*_0_ away, as shown in Figure 1. A corresponding acoustic receiver is placed in the specular reflection direction, also at a radial distance *r*_0_ from the target area on the ground. Some sound will travel a shorter direct horizontal path of length *r*_0_sin*θ* from transmitter to receiver, as shown. The receiver subtends an angle 2Δ*θ* at the transmitter.

Multiple transmitter–receiver pairs can be placed in this way around the circle at radius *r*_0_, allowing for sensing at multiple angles of incidence *θ*.

### 2.2. Selection of the Frequency of the Transmitted Acoustic Signal

The plane wave reflection coefficient, *R*, of a smooth, plane, porous surface, reduces to the asymptotic form:(1)$$R=\frac{\alpha_{\infty }\cos\theta -\phi \sqrt{\alpha_{\infty }^{2}-\sin^{2}\theta }}{\alpha_{\infty }\cos\theta +\phi \sqrt{\alpha_{\infty }^{2}-\sin^{2}\theta }}$$
providing the transmitted acoustic frequency is *f_T_* >> *f_c_*, where
(2)$$f_{c}=\frac{\phi \sigma }{2\pi \alpha_{\infty }\rho_{0}}$$
(see [13]). It is attractive to operate in this high frequency regime because *R* depends on only two soil properties, tortuosity *α*_∞_ and porosity *ϕ*. The physical parameters in (2) also include the flow resistivity *σ*, and air density *ρ*_0_. Typical values for grasslands quoted by [14] are *α*_∞_ = 1.35, *ϕ* = 0.65, and *σ* = 1 × 10^5^ Pa s m^−2^. The air density at 15 °C is *ρ*_0_ = 1.2 kg m^−3^, giving a critical frequency *f_c_* around 6 kHz. 

Derivation of (1) gives the reflectivity due to a sharp interface but allowing for the penetration of sound within the soil pores. Estimation of porosity deeper within the soil is not possible with this method. However, an indication of the sensed layer depth can be gauged. The Zwikker–Kosten model for the complex wavenumber in a porous material [14] gives
(3)$$k_{g}=k\sqrt{c_{s}+i\frac{\alpha_{\infty }}{2}\frac{f_{c}}{f_{T}}}$$
where *k* is the wavenumber in air and *c_s_* is a pore structure constant. The exponential decay of the ultrasound within the soil depends on the imaginary part of this complex wavenumber giving, for *f_T_* >> *f_c_*, an attenuation coefficient, or effective penetration depth:(4)$$\eta =\frac{2cc_{s}^{\frac{1}{2}}}{\pi f_{c}\alpha_{\infty }\cos\theta }$$
where *c* is the speed of sound in air. Using typical values of *c* = 340 m s^−1^, *c_s_* = 3, *α*_∞_ = 1.35, and *f_c_* = 6 kHz, *η* = 25 mm at *θ* = 20° and *η* = 46 mm at *θ* = 60°. This suggests that ultrasound of frequency 25 kHz is likely to penetrate a few tens of mm.

The partial filling of pores with water will reduce the porosity estimated acoustically. This will likely be a problem for the farm-scale acoustic classification of soil porosity following heavy rain or an irrigation event, although the upper few tens of mm will likely dry relatively rapidly (see also [15] as an indication of water content variation with depth). 

In practice, natural soil surfaces have a reduced specular reflection because sound is scattered away from the main beam by the rough surface. The scattering depends on the angle of incidence *θ*, the angle into which sound is scattered, wavenumber *k*, standard deviation of surface height *σ_h_*, and roughness horizontal correlation. At sufficiently high frequencies, the plane wave reflection coefficient in the specular reflection direction is reduced by the factor
(5)$$e^{-\frac{1}{2}g^{2}}$$
where
(6)$$g=2k\sigma_{h}\cos\theta$$
is the Rayleigh roughness parameter [16]. Surface height variations *σ_h_* are typically 5 to20 mm for pasture fields [17].

Natural soils often have a vegetative cover. Geometric scattering of sound by pasture blades is an approximation which assumes an acoustic frequency *f_T_* >> *f_V_*, where
(7)$$f_{V}=\frac{c}{2\pi w}$$
with *w* being the width of a pasture blade. A typical range for *w* is 2 to 6 mm [18]. The values of the critical frequencies *f_c_* and *f_V_* are shown in Figure 2 together with the variation with frequency of the roughness reduction factor given in (5).

These limits suggest use of a low ultrasound frequency. Low ultrasound, rather than audible sound, is attractive because noise from agricultural machinery will be minimized. Typical center frequencies of readily available ultrasound transmitters are 25 kHz and 40 kHz. However, there is poor penetration through the pasture of ultrasound of frequencies around 40 kHz at normal incidence [19], consistent with Figure 2, so an operating frequency of 25 kHz is chosen.

### 2.3. Selection of the Pulse Duration

In addition to the reflected ray path of length 2*r*_0_, Figure 1 shows a direct ray path of length 2*r*_0_sin*θ*. Pulses arriving at the receiver from these two paths should be separated in time so that the reflected signal can be analyzed. The time difference determines the minimum pulse duration, given by
(8)$$\tau =\frac{2r_{0}}{c}\left(1-\sin\theta \right).$$

The transmission is a sinusoid pulse at frequency *f_T_* shaped by a Hann window [20]
(9)$$V=\frac{1}{2}V_{max}\sin\left(2\pi f_{T}t\right)\left[1-\cos\left(2\pi \frac{t}{\tau }\right)\right]$$
where *V* is the voltage driving the transmitter elements, *V_max_* is the peak voltage, *τ* is the pulse duration, and *t* is time. Examples of signals received from soil samples are shown in Figure 3 for *f_T_* = 25 kHz, *r*_0_ = 1000 mm, *θ* = 30° and 52°. The predicted onset of the direct pulse arrival is shown by a green line and for the reflected pulse arrival by a blue line. For *θ* = 30°, there is also a small signal, which peaks at *t* = 5.7 ms, arising from a reflection off the frame holding the transmitter and receiver arrays. The signal does not immediately return to zero volts at the expected end of the pulse due to “ringing” of the high-Q transmitter elements.

The predicted number of cycles within the transmitted pulse is *N_c_* = *τf_T_* = 74 at *θ* = 30° and *N_c_* = *τf_T_* = 31 at *θ* = 52°. Random white amplitude noise will be inversely proportional to the number of samples, and therefore, to the number of cycles. This means it is sensible to choose pulse lengths up to the duration predicted by (6), rather than select a uniform short pulse length at all angles.

### 2.4. Selection of Transmitter and Receiver Diameters

The signal level is improved by using multiple elements in a circular planar array for the transmitter. A starting point for selecting the diameter of this array is the minimum angular spacing of transmitter array units around a circle of radius *r*_0_. For *r*_0_ = 1000 mm, a 5° (0.088 radian) spacing would be possible if the radius of each array enclosure is 0.088*r*_0_/2 = 44 mm. A circuit board diameter of 100 mm is chosen, with transmitter and receiver elements at a maximum distance of 38 mm from the board center.

Providing enough transmitter elements are within a circle of radius *a* on the transmitter array circuit board, the transmitter acts like a circular source of sound, producing a far-field Airy angular acoustic pressure pattern
(10)$$p=p_{0}\;2\frac{J_{1}\left(x\right)}{x}$$
where
(11)x=kasin(Δθ),
Δ*θ* is the angle with respect to the beam axis, *p*_0_ is the pressure on the axis at a given distance from the transmitter, and *J*_1_ is the Bessel function of the first kind [21].

From Figure 1, and in the far field, the receiver subtends at angle 2Δ*θ* = 2*a*/(2*r*_0_) at the transmitter, assuming the receiver is also of radius *a*. In (10) *x = ka*Δ*θ* = *ka*^2^/(2*r*_0_). For reception from a perfectly reflecting surface, the pressure amplitude at the rim of the receiver will be a factor:(12)$$2\frac{J_{1}\left(\frac{ka^{2}}{2r_{0}}\right)}{\frac{ka^{2}}{2r_{0}}}$$
smaller than the pressure amplitude at the center of the receiver array. As *a* increases, the transmitted beam becomes narrower and the collecting diameter becomes bigger, giving a larger variation in pressure amplitude across the receiver. Figure 4 shows this variation as a function of array radius *a*. A larger variation means that the sound intensity variation on the ground is larger in the “footprint” region to which the receiver will respond. The footprint is an ellipse of semi-minor axis *r*_0_Δ*θ* = *a*/2 and semi-major axis *a*/(2cos*θ*), which has an area of
(13)$$A_{g}=\frac{\pi a^{2}}{4\cos\theta }.$$

If *a* = 38 mm, the footprint semi axes are of length 19 mm and 20 mm, with a footprint area of *A_g_* = 1190 mm^2^ for *θ* = 17°, and semiaxes of length 19 mm and 31 mm, with area *A_g_* = 1840 mm^2^ when *θ* = 52°. This is a small footprint area which is a compromise allowing for multiple compact array packages.

### 2.5. Selection of the Angle of Incidence

From (1), multiple angles of incidence *θ* will give multiple readings of plane wave reflection coefficients *R*. This provides a set of equations from which the soil parameter tortuosity *α*_∞_ and porosity *ϕ* can be deduced, as well as surface roughness and vegetative loss. Figure 5 shows the variation of *R* with *θ* for one set of values of *α*_∞_ and *ϕ*, without surface roughness and vegetative losses. Equation (1) predicts *R* = 0 at an angle of incidence
(14)$$\theta_{max}=\tan^{-1}\left(\sqrt{\frac{\alpha_{\infty }}{\alpha_{\infty }-1}\left[\frac{\alpha_{\infty }}{\phi^{2}}-1\right]}\right).$$

A 180° change of phase also occurs for *θ* > *θ_max_*. For the examples plotted in Figure 5, this occurs around 70° when *α*_∞_ = 1.35 and *ϕ* = 0.65 and around 60° when *α*_∞_ = 1.35 and *ϕ* = 0.85. The prototype design has *M* = 4 transmitter–receiver pairs at *θ* = 17°, 30°, 40°, and 52°.

### 2.6. Transmitter and Receiver Arrays

PROWAVE Air Ultrasonic Ceramic Transducers 250ST/R160 prowave.com.tw (accessed on 30 March 2024) were chosen as the transmitting elements. Based on the device specifications, these have a transmission intensity peak near 25 kHz as shown in Figure 6 and a polar response shown in Figure 7.

The specified SPL at 25 kHz is a minimum of *SPL_spec_* = 117 dB, where 0 dB is equivalent to a rms sound pressure of *p_ref_* = 20 μPa measured at a distance of *r_spec_* = 0.3 m when driven by a sinusoidal signal of *V_spec_* = 10 Vrms. The acoustic pressure sensitivity to the driving voltage is therefore
(15)$${\frac{dp}{dV}|}_{t}=\frac{p_{ref}}{V_{spec}}10^{\frac{SPL_{spec}}{20}}$$

When a single 250ST160 transmitter element is driven by a sinusoidal pulse of rms voltage *V_t_*, a sinusoidal acoustic pressure *p_t_* is produced at the ground, which is at a distance *r*_0_ from the transmitter, according to
(16)$$p_{t}=V_{t}\frac{r_{spec}}{r_{0}}{\frac{dp}{dV}|}_{t}.$$

For example, if *V_t_* = 6 V and *r*_0_ = 1 m, *p_t_* = 2.5 Pa at the ground. Some limitations are that the maximum driving voltage is 20 Vrms and the capacitance is 2.4 nF.

The 250ST/R160 can also be used to receive ultrasound but the need for a transmit/receive switch and the fact that there are more sensitive microphones available led to choosing a WM-61A Omnidirectional Back Electret Condenser Microphone Cartridge https://micronic.co.uk/products/panasonic-wm-61a-omni-directional-mini-electret-condenser-microphone-capsule (accessed on 30 March 2024). The output of this microphone is only specified for frequencies less than 20 kHz (they are intended for audio) but past experience has shown that they work well beyond 50 kHz [19]. Sensitivity is *dB_m_* = −35 dB where 0 dB means *V_m_* = 1 Vrms output for an acoustic pressure at the microphone of *p_m_* = 1 Pa. 

The voltage *V_r_* produced by a WM-61A receiver element for a given acoustic pressure *p_r_* at its face is
(17)$$V_{r}=p_{r}{\frac{dV}{dp}|}_{r}=V_{m}\frac{p_{r}}{p_{m}}10^{\frac{dB_{m}}{20}}.$$

For these microphones, the signal-to-noise voltage ratio (SNR) for self-noise is more than 62 dB, which means that the noise voltage from this device is less than *V_m_*10^−*dBm*/20^ = 0.8 mVrms.

The voltage output *V_r_* from a single receiver element due to reflection from a ground surface of plane wave reflection coefficient *R* is therefore related to the voltage *V_t_* driving a single transmitter element via
(18)$$\frac{V_{r}}{V_{t}}=R\frac{r_{spec}}{2r_{0}}{\frac{dp}{dV}|}_{t}{\frac{dV}{dp}|}_{r}=GR$$

Assuming *r*_0_ = 1 m, *R* = 0.5, and *V_t_* = 10 V, *V_r_* = 10^−7^ V. A 16-bit ADC with a maximum input voltage of 10 V will have its least significant bit representing 15 mV, so a considerable circuit gain is required. This can be achieved by using multiple transmitter elements and multiple receiver elements, as well as using a receiver amplifier.

The transmitter design shown in Figure 8 uses 9 of the 250ST160 transmitter elements. The outer 6 of these are arranged on a circle of radius 38 mm and the inner 3 are on a circle of radius 18 mm. Each transmitter element has an active transmitting area of diameter 14 mm, so the effective transmitter array radius is 45 mm. 

The receiver design shown in Figure 9 uses 9 of the WM-61A microphones, with the outer 6 on a circle of radius 38 mm and the inner 3 on a circle of radius 18 mm, as for the transmitter. Each array has a central laser diode for accurate pointing alignment.

The 9 transmitter elements are driven in parallel from an operational amplifier via a npn/pnp transistor driver pair within the feedback loop, since the 250ST160 elements are largely capacitive (2.4 nF each) and driving a total capacitive load of 22 nF needs to be considered. Power is supplied from two 9V batteries, although the circuit will operate from ±18 V so the batteries could be doubled up to provide a more intense signal.

Each WM-61A microphone is buffered internally with a FET and needs to be supplied with current through a load resistor. The small-signal voltage across each load resistor is capacitively coupled into the summing junction of an operational amplifier, and the 9 signals are added. The receiver circuit also contains a band-pass filter with gain.

### 2.7. Data Acquisition

Two Data Translation DT9832 data acquisition units www.mccdaq.com (accessed on 30 March 2024) are used to produce pulses and receive the echo signals. Each unit has two ±10 V DAC outputs, allowing connection to 4 transmitter units. The DT9832 has a 10 V amplitude 16 bit DAC and ADC and the sampling rate for both is set at 200 kHz. A Hann-windowed pulse is transmitted, and the echo signal is received simultaneously. Measurements show a 0.5 μs delay, equivalent to a path length of less than 0.2 mm, which is acceptable. The maximum and minimum signal levels are examined automatically to check that there is no clipping in the receiver. If clipping occurs, the reflection data are discarded, and the transmitter drive voltage is lowered. The received signals are displayed in real time, leading to around 30 pulses per second, depending on the software and computer used. This overall loop is repeated 32 times for each angle of incidence. An operational design will most likely transmit simultaneously on all transmitter arrays.

### 2.8. System Gain and Acoustic Beam Shape

If the ground was a perfect reflector, the received signal would be the same as that from a direct path from a transmitter facing the receiver at a distance of 2*r*_0_. In that case, the overall system gain *G* can be found from (16) with *R* = 1. Measurements were conducted in the laboratory by directing a transmitter toward a receiver separated by a distance of 2*r*_0_ = 2000 mm. The beam shape is measured by rotating the transmitter in small angular increments. Results are shown in Figure 10.

These system gain results can be summarized as
(19)$$G\left(\Delta \theta \right)=2G_{0}\frac{J_{1}\left(x\right)}{x}$$
where *G*_0_ = 9.1 ± 0.04 and
(20)x=kasin(Δθ)
with *a* = 51 mm and *f_T_* = 25 kHz. The effective array radius of 51 mm is slightly larger than the circuit board, possibly because of some reflections from the array casing.

The half-power value of *x* is 1.616 or Δ*θ* = 3.9°. Alternatively, *G*(Δ*θ*) may be approximated by a Gaussian of standard deviation
(21)$$\sigma_{\Delta \theta }\approx \frac{\sqrt{2}}{ka}$$
giving *σ*_Δ*θ*_ = 3.4° [22].

The above calibration allows measurements to be made from which the plane wave reflectivity is
(22)$$R=\frac{1}{G_{0}}\frac{V_{r}}{V_{t}}.$$

A Monte Carlo simulation was performed with random normal fluctuations in *G* having a standard deviation of 0.04. and with *α*_∞_ =1.35, *ϕ* = 0.7, and *σ_h_*.= 0. The uncertainty in the estimated *ϕ* was 0.13%.

Random measurement noise is from either electronic noise in the transmitter and receiver electronics, or spurious ultrasonic noise received along with the reflected signal. Random measurement noise in voltage *V_r_* causes random errors in the estimated plane wave reflection coefficients *R_m_* at each angle *θ_m_*, according to (22). These random errors affect the parameters estimated according to (23). The peak voltage of the received pulse is used to estimate *R_m_*. An estimate of the effect of the voltage noise was made by comparing voltages recorded at the same phase of a 25 kHz cycle for 3 cycles at the top of the recorded pulse, from measurements made over bare agricultural ground. The standard deviation of *V_r_* found in this way was 0.4 mV, whereas the peak *V_r_* is typically 6 V. The relative rms error in peak voltage due to random noise is therefore typically less than 0.01%. The low level of measurement noise is also evident in Figure 3. A Monte Carlo simulation was also performed using the measured 0.4 mV fluctuations to show that the effect on estimated porosity was less than 0.01%. This shows that random measurement noise has a negligible effect on the estimation of porosity. 

The most likely systematic measurement error is an inaccurate setting of the angle of incidence *θ*. This angle was built into a frame holding the transmitters and receivers and all transmitters and receivers had a laser diode beam which showed the acoustic beam center on the ground, as shown in Figure 11. The setup errors should therefore be small. A Monte Carlo simulation shows that a 2.5° standard error in angle settings results in a 0.028 error in porosity estimation. Errors of this size are not significant for interpreting porosity data, but clearly pointing out the transmitters and receivers needs to be carefully carried out.

## 3. Results

The objective is to estimate the porosity of pasture-covered agricultural soils which have a naturally rough surface. Interpretation of measurements requires using a composite of three models: a model for reflection of high-frequency sound from a porous surface; a model for scattering losses of ultrasound by a random rough surface; and a model for scattering losses of ultrasound by pasture. The first of these, described by (1), and the second, described by (5), are well-established. No prior work exists for the high-frequency scattering of sound from pasture. Development of a suitable model would need to consider the random orientation of pasture blades and shapes and would involve considerable laboratory and field testing. This work will be left for further investigation. 

Operating over bare rough soil involves finding the set of soil parameters *ϕ*, *α*_∞_ and *σ_h_* which best match measurements of the received signal *V_r_*. Measurements *V_r_* are made and *R_m_* found from (22), at known incident angles *θ_m_*, *m* = 1, 2,…, *M*. The sum of the squared residuals: (23)$$\chi^{2}=\overset{M}{\underset{m=1}{\sum }}{\left[R_{m}-\frac{\alpha_{\infty }\cos\theta_{m}-\phi \sqrt{\alpha_{\infty }^{2}-\sin^{2}\theta_{m}}}{\alpha_{\infty }\cos\theta_{m}+\phi \sqrt{\alpha_{\infty }^{2}-\sin^{2}\theta_{m}}}e^{-\frac{1}{2}{\left(2k\sigma_{h}\cos\theta_{m}\right)}^{2}}\right]}^{2}$$
is minimized to give the best estimates, *ϕ*, *α*_∞_ and *σ_h_*, in the least squares sense, of the soil physical parameters. The minimum χ^2^ can be found through a simple search over a grid of physically realistic ranges of the soil parameters *ϕ*, *α*_∞_ and *σ_h_*. The *M* = 4 incident angles were 17°, 30°, 40°, and 52°. The effect of changing grid steps for each of *ϕ*, *α*_∞,_ and *σ_h_* was tested to help ensure that multiple minima did not exist.

Five soil samples labelled from 1 to 5 were tested in the laboratory. Figure 11 shows the experimental configuration in the laboratory, which was also the configuration used in the field. Two soil samples, labelled 6 and 7, were also tested in situ in the field with the overlying pasture removed. These field samples were then cut out and placed in pans. Both the lab samples and the bare field samples were oven dried (24 h at 105 °C) and weighed. The samples analyzed in the laboratory are shown in Figure 12 (photographs were not taken of the field samples). Note that these samples are less rough than is typical of agricultural soils [23,24]. 

After the acoustic test, the samples were saturated for about 4 days and weighed again. Porosity was determined using the gravimetric method [25] by comparing the saturated weight and oven-dried weight of the samples, giving porosity estimates of *ϕ_gravimetric_* = 0.55, 0.50, 0.61, 0.60, 0.58, 0.52, and 0.54 for samples 1 to 7 respectively. The porosities estimated from ultrasonic measurements were *ϕ_ultrasonic_* = 0.52, 0.52, 0.65, 0.65, 0.57, 0.59, and 0.50 for samples from 1 to 7, respectively, as shown in Figure 13.

The square root of the mean of the squares of the differences between *ϕ_ultrasonic_* and *ϕ_gravimetric_* (the rmse) is 0.04, or around 8%. These differences are due to a combination of factors such as soil surface variations across the footprint, and model over-simplification. It is likely that such factors will also affect the estimates of tortuosity *α*_∞_ and roughness *σ_h_* in a similar way. The variation of χ^2^ above its minimum at *ϕ_ultrasonic_* ± 0.04 gives a guide to the dependence on the other parameters, giving the standard deviation of estimates of *α*_∞_ as 0.08 and of *σ_h_* as 0.30 mm. The estimates of the rms surface height, *σ_h_*, for the soil samples from 1 to 4 are 0.00, 1.23, 1.53, and 2.10 mm, agreeing with the visual assessment of roughness in Figure 12. For the fifth soil sample, an optical scan of the surface height was performed [26], giving an rms surface height variation of *σ_h_* = 1.23 mm, and that was estimated using ultrasound at 1.62 mm. The optical estimate can also be expected to contain uncertainties, so the agreement is not unreasonable. These results are shown in Table 1.

## 4. Discussion

It is relatively straightforward to use reflections of ultrasound to measure the porosity of samples of materials in the laboratory [27,28,29], but the intent of the current work is to design a readily useable technology for routine paddock-scale porosity estimation. This creates some challenges because the design must comprise a compact low-power rigid structure and allow for pasture cover and surface roughness. 

The high-frequency regime for acoustic interactions with a porous material involves the porosity *ϕ*, the tortuosity *α*_∞_, and the angle of incidence *θ* [13]. There is no dependence on frequency, so measurements must be conducted at two or more angles of incidence *θ* to obtain two or more equations in the unknown *ϕ* and *α*_∞_. A system requiring a single transmitter/receiver pair to be moved from one angle to another is not viable for rapid sampling. Therefore, for specular reflections, at least two transmitter/reflector pairs are required. Ideally, these are placed at identical distances from the ground so that the power reflected from a perfectly reflecting surface would be the same.

To operate in the high-frequency regime for porosity, the frequency needs to be well above 6 kHz. In the low-ultrasound frequency range, the loss of the reflected signal due to scattering by surface roughness also takes a simple form [16]. The result is that the combination of porosity effects and roughness effects can be handled using a single 25 kHz frequency with measurements at three or more angles.

Penetration of pasture also appears to be adequate in the low-ultrasound frequency range below about 30 kHz [19], but a model for scattering of ultrasound by pasture is required to interpret the reflections, which will be treated in a future article. 

Given available transmitter and receiver elements, it is desirable to include multiple transmitter elements into a transmitter array and multiple receiver elements into a microphone array. The array design is a pragmatic choice based on compactness of the overall system and on providing sufficient flexibility in choosing angles of incidence. The result is arrays of diameters of 100 mm, although this is a little arbitrary.

Pulsed ultrasound is used so that the reflected sound can be separated from sound travelling directly through the air from transmitter to receiver. Because of the directionality achieved by an array, and because of the directionality of the individual acoustic components, the intensity of the direct sound is not large, but would interfere significantly with the reflected signal.

Calibration of the transmitter/receiver pairs is discussed. It is straightforward to point a transmitter array at a receiver array and obtain a combined transmitter/receiver gain, which is what is required in practice. The acoustic beam shape has also been measured for confirmation of expectations (this is not required for data interpretation and in deriving porosity). 

The design has been tested on a limited number of soil samples in the laboratory and on bare agricultural soils. The result is that the average porosity estimated for each sample was within around 0.04 of the independently measured porosity value. This level of variation is consistent with uncertainties in setting the angle of incidence. Assumptions made in modelling the interaction of ultrasound with the rough surface no doubt also contribute. The 0.04 variation in porosity estimation is a very encouraging result, but the methodology needs to be tested on a much wider range of soil samples and with a range of pasture biomass cover. No allowance is currently made for larger scale “pugging” by animals, or when the footprint contains a larger rock, and soil pores partially filled with water will likely give a false impression of the overall soil porosity. It is possible that roughness outliers, such as pugging or rocks, could be detected by using receivers at multiple angles for each transmission, so that the angular symmetry of scattering by roughness elements is evaluated, but this idea has not yet been tested. In order to provide further confidence in this methodology, future work would include testing on a wider range of soils including, for example, sandy soils. Extending to vegetation-covered soils is clearly a priority and would also include evaluation of environmental factors such as wet vegetation and experience around the partial filling of pores with water.

The measurements described were made with the transmitters and receivers fixed in space. For eventual operational applications the sensor platform will be moving. Even at a typical walking speed of 4.5 km h^−1^ (1.25 m s^−1^), successive measurements at around 30 ms intervals will have footprints which do not overlap. This means that regressions will need to be performed on each reflected pulse, and then estimated porosity values averaged if required. Given the low random noise levels, this should not present a problem.

## 5. Conclusions

Directional ultrasonic transmitter and receiver arrays, operating at 25 kHz, have been designed to quantify soil porosity. The method is based on simultaneous, multi-angle non-contact measurements of ultrasonic reflections from the soil, allowing soil porosity measurements in real time. A number of constraints are described, relating to the ultrasound scattering and to the required geometry of the sensor array placements. The resulting design is compact and relatively inexpensive, and capable of being mounted on a small farm vehicle. This results in a soil porosity monitoring method which has promising potential for operational use at paddock scales. The method is novel, there currently being no other non-invasive (non-contact) proximal sensing methods which directly allow for the estimation of soil porosity.

## Figures and Tables

**Figure 1 sensors-24-02266-f001:**
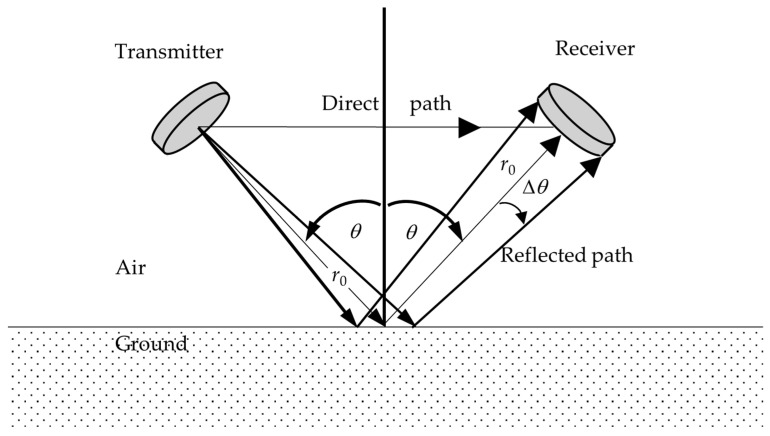
The geometry of the placement of transmitter and receiver pairs. The transmitter unit and receiver unit are both a distance *r*_0_ from the target area on the ground. The angle of incidence is *θ* and the receiver subtends an angle 2Δ*θ* at the transmitter. A horizontal direct path of length 2*r*_0_sin*θ* is shown.

**Figure 2 sensors-24-02266-f002:**
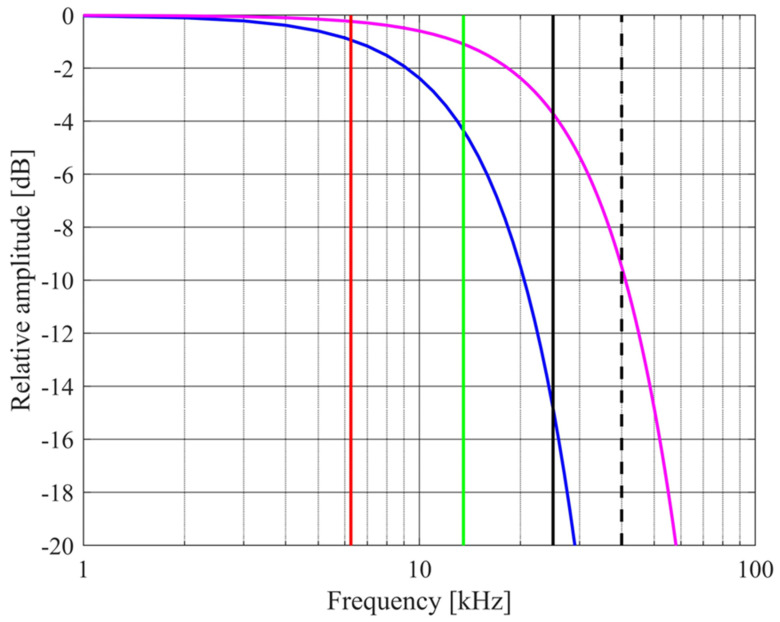
Frequency dependencies: *f_c_* with *α*_∞_ = 1.35, *ϕ* = 0.65, and *σ* = 1 × 10^5^ Pa s m^−2^ (red line); *f_V_* with *w* = 4 mm (green line); exp(−^2^/2) with *σ_h_* = 2 mm and *θ* = 0° (blue curve); and exp(−*g*^2^/2) with *σ_h_* = 2 mm and *θ* = 60° (magenta curve). Central frequencies of typical transmitter elements are 25 kHz (solid black line) and 40 kHz (dashed black line).

**Figure 3 sensors-24-02266-f003:**
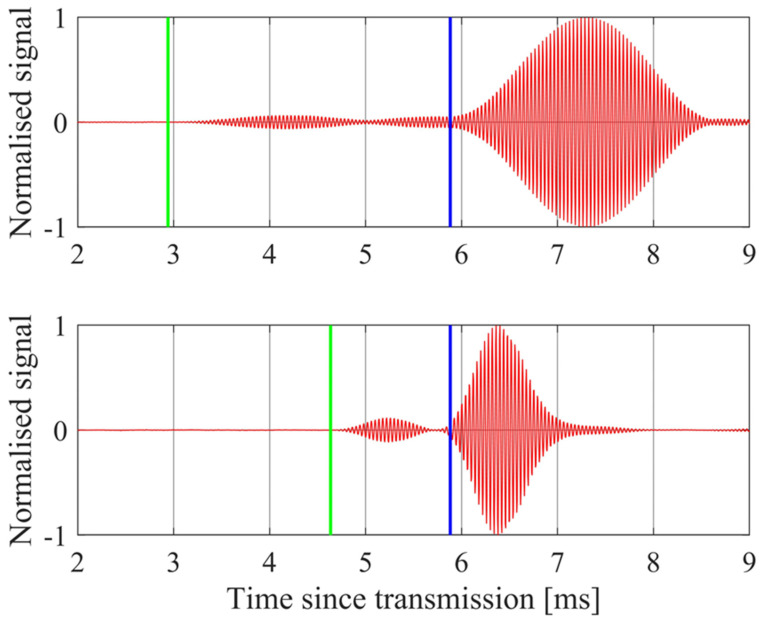
The normalized measured signal (red curve) from a soil sample at *θ* = 30° (upper plot) and at *θ* = 52° (lower plot), together with the predicted onset of the direct pulse arrivals (green lines) and the reflected pulse arrivals (blue lines). A small reflection from the supporting frame is also evident ahead of the ground reflection in the *θ* = 30° case.

**Figure 4 sensors-24-02266-f004:**
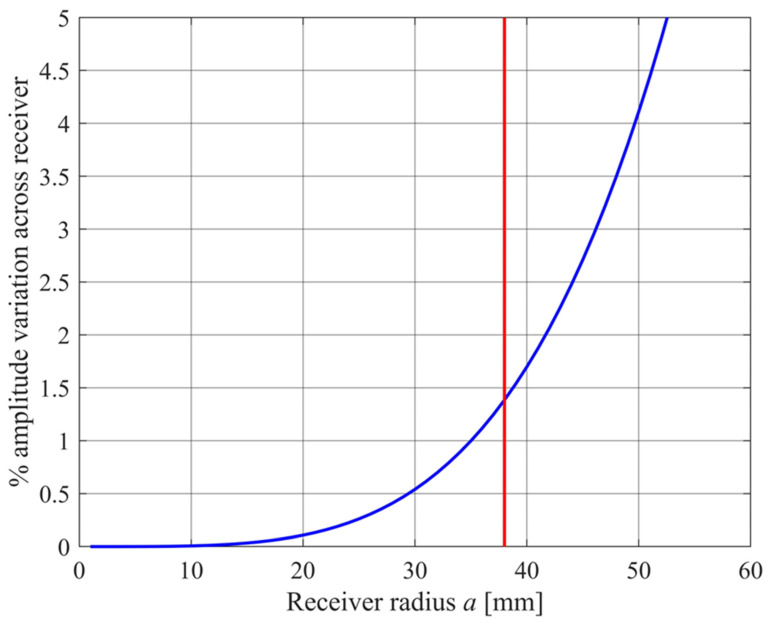
The decrease in amplitude across a receiver of radius *a*, also given a transmitter of radius *a* (blue curve) and at the chosen array radius of 38 mm (red line). The transmitted frequency is *f_T_* = 25 kHz and the range is 2*r*_0_ = 2000 mm.

**Figure 5 sensors-24-02266-f005:**
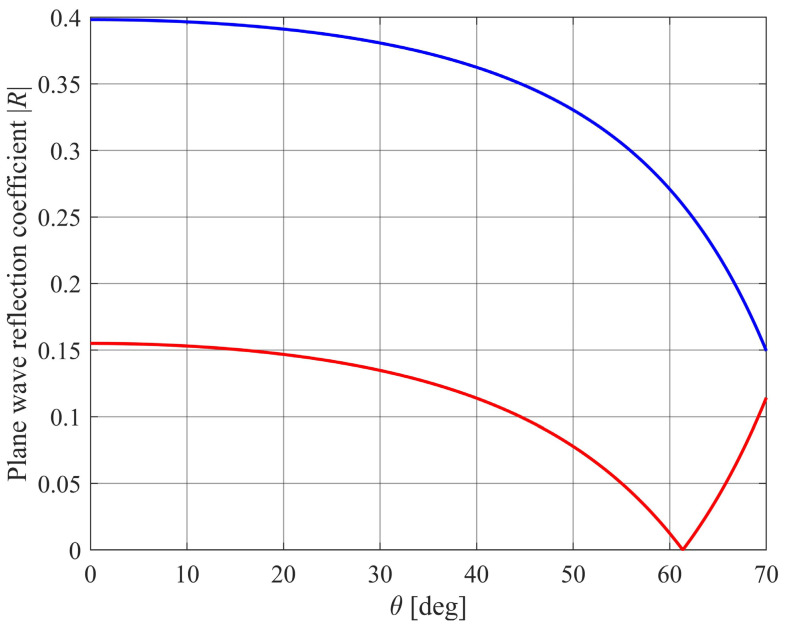
Predicted plane wave reflection coefficient *R* for a porous surface with *α*_∞_ = 1.35, *ϕ* = 0.50 (blue curve) and for *α*_∞_ = 1.35, *ϕ* = 0.85 (red curve).

**Figure 6 sensors-24-02266-f006:**
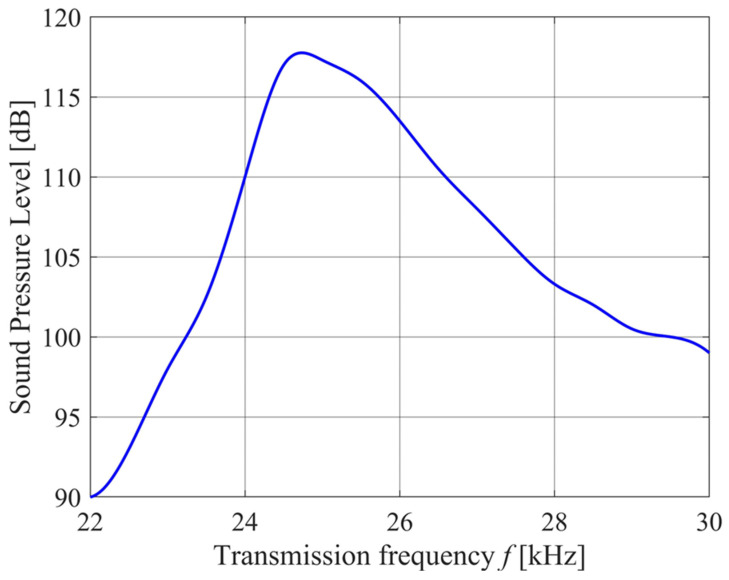
The SPL produced by a single 250ST/R160 transmitter element at a distance of *r_spec_* = 0.3 m when driven by a sinusoidal signal of *V_spec_* = 10Vrms.

**Figure 7 sensors-24-02266-f007:**
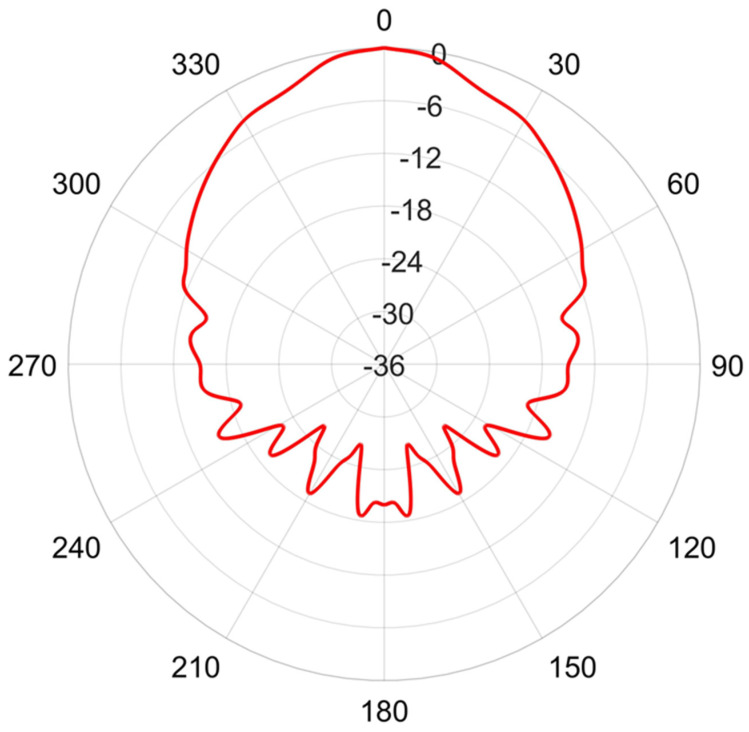
The normalized polar response, in dB, of a single 250ST160 transmitter element at 25 kHz.

**Figure 8 sensors-24-02266-f008:**
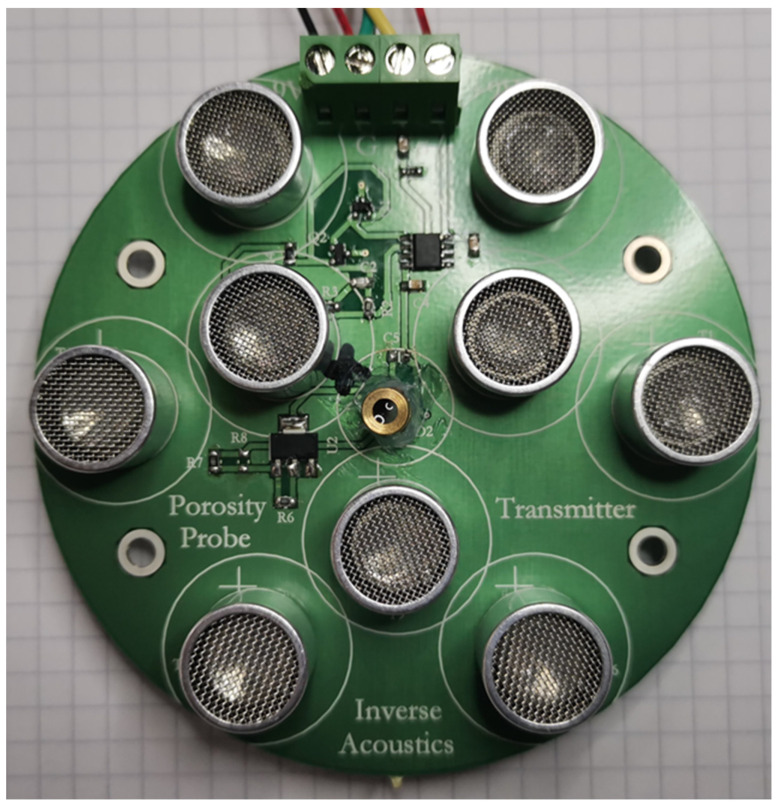
The transmitter circuit board. The large circular items are the transmitter elements.

**Figure 9 sensors-24-02266-f009:**
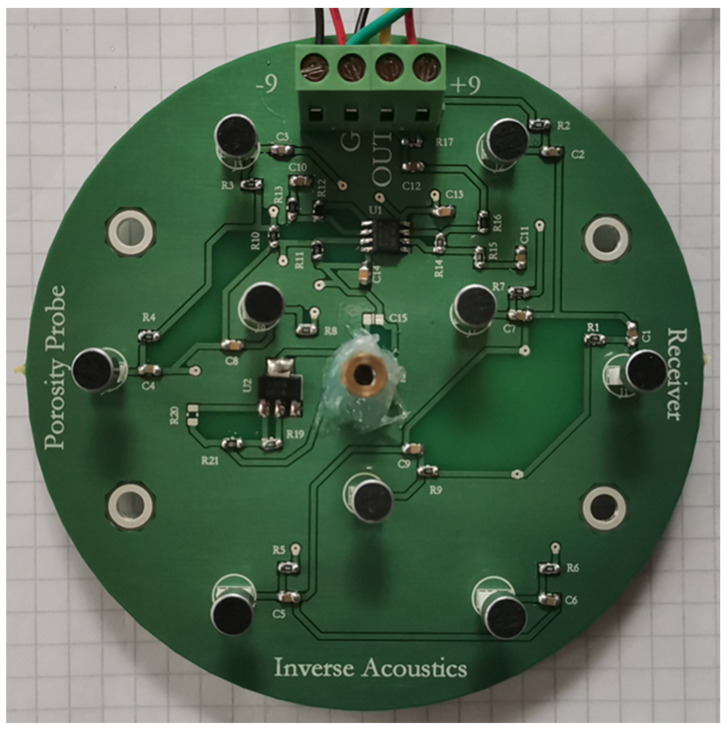
The receiver circuit board. The small circular black items are the ultrasonic receiver elements and the central brass tube is a laser diode.

**Figure 10 sensors-24-02266-f010:**
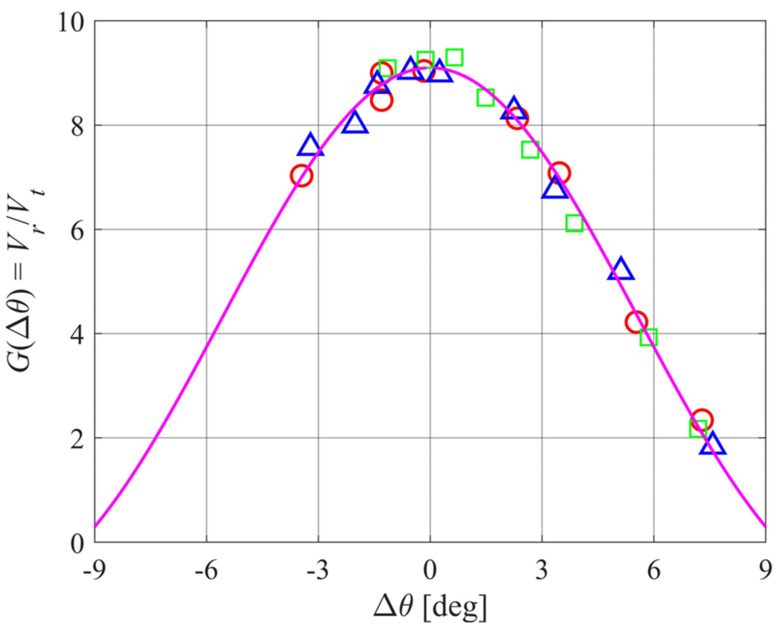
Results of measured system gain *G*(Δ*θ*) for 3 transmitter–receiver pairs (red circles, green squares, and blue triangles). Also shown is the Airy pattern (8) with *a* = 51 mm and scale factor *G*_0_ = 9.1 (magenta curve).

**Figure 11 sensors-24-02266-f011:**
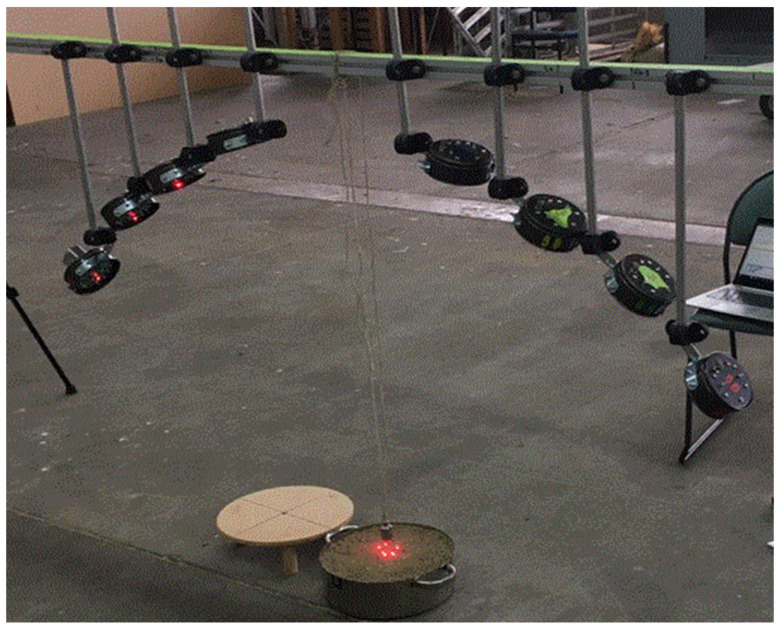
Setup for laboraory testing of soil samples.

**Figure 12 sensors-24-02266-f012:**
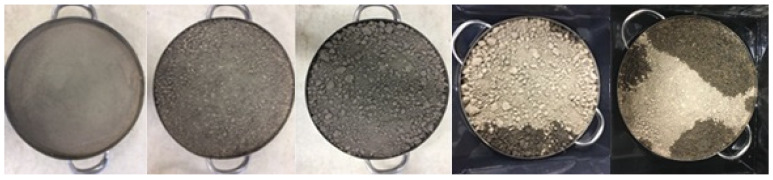
Soil samples 1 to 5 (numbering from the left).

**Figure 13 sensors-24-02266-f013:**
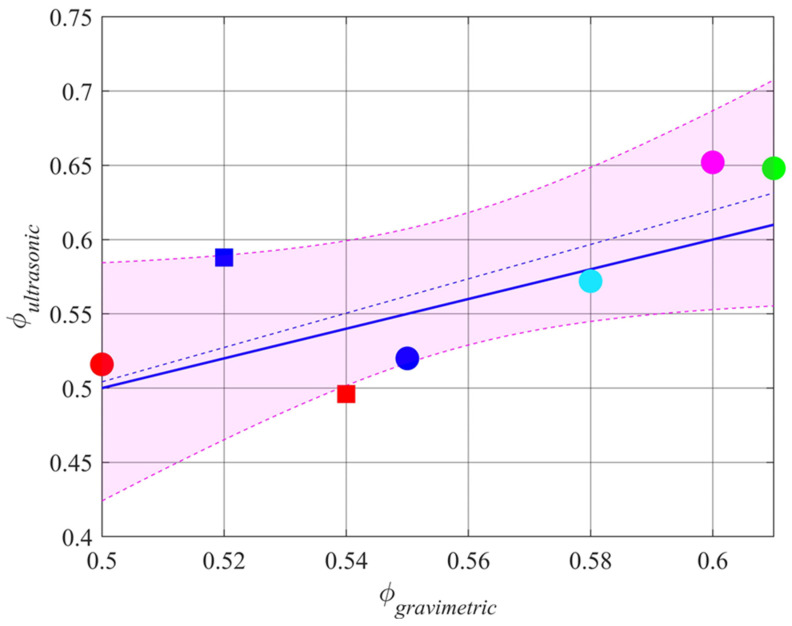
Results of soil porosity estimation compared with the porosity values independently estimated from the gravimetric method. The results for laboratory soil samples are shown as circles (soil 1 blue, soil 2 red, soil 3 green, soil 4 magenta, soil 5 cyan), those from bare field samples by squares (soil 6 blue, soil 7 red). The least squares regression is shown as a dashed blue line and 1:1 as a solid blue line. The magenta shaded area is the 95% confidence bounds.

**Table 1 sensors-24-02266-t001:** Porosity estimated from ultrasonic reflections compared with porosity estimated by the gravimetric method. Soils from 1 to 5 are laboratory samples, and soils 6 and 7 are from a field site after pasture cover was removed. The estimated rms surface heights and tortuosity values are also given.

Soil	*ϕ_ultrasound_*	*ϕ_gravimetric_*	*σ_h_* (mm)	*α*_∞,_ *_ultrasonic_*
1	0.52	0.55	0.00	1.99
2	0.52	0.50	1.23	1.93
3	0.65	0.61	1.53	1.30
4	0.65	0.60	2.10	1.22
5	0.57	0.58	1.62	1.20
6	0.59	0.52	1.44	1.96
7	0.50	0.54	1.44	1.28

## Data Availability

The datasets presented in this article are not made available because the data are part of an ongoing study involving the estimation of porosity of pasture-covered soils.

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
