# Peer review of "Design of an Ultrasound Sensing System for Estimation of the Porosity of Agricultural Soils"

_sensors, 2024, doi:10.3390/s24072266_

Round 1

Reviewer 1 Report

Comments and Suggestions for Authors

There are many aspects that need clarification before it can proceed for publication. Below, I have outlined my comments:

Comment 1

- Authors in the abstract fail to address the potential limitations or challenges associated with the proposed ultrasound sensing system. They do not discuss factors such as environmental conditions, calibration requirements, or the accuracy of porosity estimation in diverse soil types, which are critical considerations for the practical applicability of the technology.

Comment 2

- In the introduction there is a lack of discussion on the current state-of-the-art in soil porosity measurement.

Comment 3

- The authors of the paper should consider providing a more explicit explanation of the novelty and significance of their proposed ultrasound sensing system for soil porosity estimation.

Comment 4

-Figure 1 is not well explained ,Authors should give a deep explanation of this figure.

Comment 5

- In section 2.7 the authors claim that random noise has a negligible effect, the methodology used to estimate and mitigate this noise is not adequately justified or transparent.

Comment 6

- No explanation of how calibration was performed and how potential sources of error were minimized, it is difficult to assess the reliability and accuracy of the porosity estimations obtained through this system.

Comment 7

- While the authors mention testing on a limited number of soil samples, they fail to outline a clear plan for expanding the scope of testing to ensure the robustness and reliability of the system across diverse soil types and environmental conditions.

Author Response

The authors are grateful for the evident time and care the Reviewer put into their in-depth questions and suggestions. The following treats each point, also referring to the new line numbers (hopefully these will be the same line numbers once the revised manuscript reaches the Reviewer).

1) Authors in the abstract fail to address the potential limitations or challenges associated with the proposed ultrasound sensing system. They do not discuss factors such as environmental conditions, calibration requirements, or the accuracy of porosity estimation in diverse soil types, which are critical considerations for the practical applicability of the technology.

We have added the following in the Abstract (lines 26 – 28) “Although the method is applicable to all soil types, the current design has only been tested on dry, vegetation-free soils for which the sampled area does not contain large animal footprints or rocks.” This is expanded in the Discussion (lines 402-409).

2) In the introduction there is a lack of discussion on the current state-of-the-art in soil porosity measurement.

We have added a further 6 citations in the Introduction, including review articles (one of which is 2023). We also point out that none of these state-of-the-art proximal soil sensing methods provide direct, non-invasive, estimation of soil porosity, which is the goal of our current work. This is emphasised in the Conclusions (lines 421-422).

3) The authors of the paper should consider providing a more explicit explanation of the novelty and significance of their proposed ultrasound sensing system for soil porosity estimation.

You are right. We have now included a statement regarding novelty in the Abstract (lines 12-13) and also in the Conclusions (lines 421-422).

4) Figure 1 is not well explained, Authors should give a deep explanation of this figure.

Figure 1 has been simplified, since the “image receiver” may have been confusing. There is deep discussion of the distances and angles in Section 2.4.

5) In section 2.7 the authors claim that random noise has a negligible effect, the methodology used to estimate and mitigate this noise is not adequately justified or transparent.

We have expanded the discussion of random electronic and ultrasonic noise which now appears in Section 2.8 (lines 273-283).

6) No explanation of how calibration was performed and how potential sources of error were minimized, it is difficult to assess the reliability and accuracy of the porosity estimations obtained through this system.

The calibration method in Section 2.8 has been described in greater detail. Soil parameters are estimated from (23), in which Rm = Vr/(G0 Vt). Vt is known very accurately from the signal sent to the DAC. Vr has a low measurement noise as also described. Calibration is required to estimate the system gain G0. As described in the calibration in Section 2.8, G0 = 9.1 +/- 0.04. We have now done a Monte Carlo analysis of the effect of this calibration uncertainty on the estimation of porosity and found 0.13% random error in porosity (lines 271-272).

7) While the authors mention testing on a limited number of soil samples, they fail to outline a clear plan for expanding the scope of testing to ensure the robustness and reliability of the system across diverse soil types and environmental conditions.

The Discussion now includes potential future work to: (1) sense the uniformity of surface roughness (to allow discarding of data with roughness outlier caused by, for example, a rock within the footprint area); (2) testing on a wider range of soils; and (3) extending to vegetation-covered soils, which is clearly a priority. Extended work on vegetated soils would also include evaluation of environmental factors such as wet vegetation and experience around partial filling of pores with water (lines 402-414).

Reviewer 2 Report

Comments and Suggestions for Authors

This paper aims to design an ultrasound sensing system for soil porosity estimation based on an established theory. Several key parameters are optimized. The results will be of interest to many within the field. Some minor revisions as described below are expected.

1. As can be seen from Figure 11, the surface roughness for soil sample 4 is rather obvious. Does such rough surface have any effect on the wave reflection coefficient measurement?

2. In this work, the soil porosity is measured via wave reflection coefficient at the “air / soil” interface. Does this mean that this method is only able to measure surface porosity? How about internal porosity?

3. If the measured soil porosity is sensitive to small deviation of the incidence angle θ?

4. A photograph showing how the transmitter and receiver circuit boards are arranged and work during measurement should be added for clarity.

5. In Figure 11, why not provide photographs of all seven samples? Scalebar should also be added.

Author Response

The Reviewer has raised some important points, which we believe we have addressed in the following.

1) As can be seen from Figure 11, the surface roughness for soil sample 4 is rather obvious. Does such rough surface have any effect on the wave reflection coefficient measurement?

As we describe (lines 105-106), the R decrease as the surface becomes rougher. This is due to scattering of ultrasound into wider angles so that not as much acoustic energy arrives at the receiver. This effect is quite clear in the recorded waveforms. Our measurements at the 4 angles are performed sequentially, rather than simultaneously. This avoids the possibility of a rougher surface directing some sound onto receivers other than the one corresponding to the transmitter at the same angle of incidence.

2) In this work, the soil porosity is measured via wave reflection coefficient at the “air / soil” interface. Does this mean that this method is only able to measure surface porosity? How about internal porosity?

This is a valid point since the theory treats the interface as being abrupt. We have added substantial clarification of the penetration depth of the ultrasound into the soil, while acknowledging that this is not necessarily an indication of access to internal porosity (lines 91-104).

3) Is the measured soil porosity is sensitive to small deviation of the incidence angle θ?

This is dealt with at the end of Section 2.8 when the acoustic beam shape is calculated and measured. A Monte Carlo simulation suggests the sensitivity to setup errors in angle of incidence is small, but we also designed with visible laser diode beams to allow checking alignment (lines 284-289).

4) A photograph showing how the transmitter and receiver circuit boards are arranged and work during measurement should be added for clarity.

A new figure, Figure 11, has been added, showing the experimental configuration.

5) In Figure 11, why not provide photographs of all seven samples? Scalebar should also be added.

Photos of the 5 laboratory samples are now included (now Figure 12). Unfortunately, photos were not taken of the two field soil samples. We do have a photo of the field configuration, but this configuration uses the same frame shown in Figure 12.

Reviewer 3 Report

Comments and Suggestions for Authors

The current state of the art in the processing of sensor data makes it possible to extend the range of applications for sensors. This manuscript proposes a method to estimate the porosity of agricultural soils based on ultrasound sensing system. The authors give great attention to the technical details of the developed device, as well as the mode of operation of the sensors. The results of laboratory and field experiments using porosity determination are presented and compared with those obtained using the gravimetric method.

The manuscript is well-organized, and the authors describe the proposed method comprehensively. The following comments are ones that should be addressed by the authors:

1)     A review of the current state of the art in estimating soil porosity based on different proximal sensor assessments (LiDAR, ultrasound, photographs) is missing from the study. As a review, the authors only describe traditional methods. At the same time, the suggested reference [6] is not relevant to this study, as this paper is dedicated to proximal estimation methods for grasslands and pastures, such as measuring plant height, estimating species composition and other parameters, and not to estimating soil porosity.

2)     The authors correctly point out that soil moisture will be a problem in determining porosity, but reference [9] is unlikely to confirm a constant amount of water in the soil within a single farm. The soil is a complex multiphase composite structure, which means that the parameter of soil moisture saturation also needs to be taken into account by the model. As this is the first stage of the study, it is perhaps better to state that the method so far is for dry soil.

3)     An optimization problem for the determination of three soil physical parameters is proposed in this paper. What method is used to solve the proposed optimization problem (according to the description, the solution is on a grid)? Is the surface smooth enough to guarantee the stability of the minima found?  How many incident angles were used in the experiments to construct the function of the sum of the squares of the residuals?

1)     The result of the three parameter estimates in the optimization problem for all seven experimental soils should preferably be presented in a table together with the estimates obtained by the other methods. The error in the porosity estimate is better expressed in relative rather than absolute terms.

2)     It would be desirable to provide a more detailed description of the experiment: photos of all seven soil samples, photos of the experimental setup in the laboratory and in the field. The paper describes an alternative method for estimating porosity, although the determination of alternative values for the other two parameters (tortuosity and surface height variations) is not clear. It is desirable to give an overall view of all three parameters and the relative errors of their estimation by the ultrasonic method.

Author Response

The authors are grateful to the Reviewer for their in-depth comments and questions, which we believe we have addressed below.

1) A review of the current state of the art in estimating soil porosity based on different proximal sensor assessments (LiDAR, ultrasound, photographs) is missing from the study. As a review, the authors only describe traditional methods. At the same time, the suggested reference [6] is not relevant to this study, as this paper is dedicated to proximal estimation methods for grasslands and pastures, such as measuring plant height, estimating species composition and other parameters, and not to estimating soil porosity.

Yes, reference [6] does not refer to soil sampling and this citation has been removed. We have added a further 6 citations in the Introduction, including review articles (one of which is 2023). We also point out that none of these state-of-the-art proximal soil sensing methods provide direct, non-invasive, estimation of soil porosity, which is the goal of our current work. This is also noted in the Conclusion (421-422).

2) The authors correctly point out that soil moisture will be a problem in determining porosity, but reference [9] is unlikely to confirm a constant amount of water in the soil within a single farm. The soil is a complex multiphase composite structure, which means that the parameter of soil moisture saturation also needs to be taken into account by the model. As this is the first stage of the study, it is perhaps better to state that the method so far is for dry soil.

We absolutely agree. Yes, our method applies only to those situations in which the pores, within the penetration range of the ultrasound, are not water-filled. We have now added a paragraph on penetration depth, which is likely to be only a few tens of mm. such a small depth is likely to dry out reasonably rapidly, but we will really only test this through more expensive sampling. We have also changed our reference to [9] (now [15]) to present that as just one possible example (lines 87-104).

3)  An optimization problem for the determination of three soil physical parameters is proposed in this paper. What method is used to solve the proposed optimization problem (according to the description, the solution is on a grid)? Is the surface smooth enough to guarantee the stability of the minima found?  How many incident angles were used in the experiments to construct the function of the sum of the squares of the residuals?

We have expanded the paragraph describing the minimization, since these are very valid questions. When we changed grid step sizes here was no indication of jumping between multiple alternative minima, and plots of the chi-square behaviour near minima were very smooth (lines 302-305).

4)  The result of the three parameter estimates in the optimization problem for all seven experimental soils should preferably be presented in a table together with the estimates obtained by the other methods. The error in the porosity estimate is better expressed in relative rather than absolute terms.

Results are now given in Table 1 for all 7 soil samples.

5)  It would be desirable to provide a more detailed description of the experiment: photos of all seven soil samples, photos of the experimental setup in the laboratory and in the field. The paper describes an alternative method for estimating porosity, although the determination of alternative values for the other two parameters (tortuosity and surface height variations) is not clear. It is desirable to give an overall view of all three parameters and the relative errors of their estimation by the ultrasonic method.

A new figure, Figure 11, has been added, showing the experimental configuration in the laboratory. This configuration is the same as that used in the field (we have a photo of the apparatus in the field, but the photo doesn’t add much).  Unfortunately we did not photograph the 2 soil samples from the field. We have also added Table 1 to show results for all parameters estimated for all soil samples. It is true that we don’t have any alternate method (“ground truth”) for tortuosity. But this is not evidently as important a parameter for soil health as is porosity. We do describe an alternative estimation of soil roughness height using a laser scanner, and give a comparison on one of the soil samples. We describe a method for estimating the relative errors for tortuosity and roughness (lines 351-360). The relative errors are 7% for porosity, 6% for tortuosity, and 54% for roughness height.

Round 2

Reviewer 1 Report

Comments and Suggestions for Authors

I accept the paper in its actual form.

Reviewer 3 Report

Comments and Suggestions for Authors

Thanks to the authors for specific answers, corrections and details of the experiment. The manuscript contains small typos, please proofread it carefully.